

# Regulatory role of permanent gully in runoff dissolved nitrogen and phosphorus transport across rainfall types

Zhuoxin Chen[1], Mingming Guo[1], Lixin Wang[1,3,5], Xin Liu[1], Jinshi Jian[2], Qiang Chen[4], Xingyi Zhang[1]

[1] State Key Laboratory of Black Soils Conservation and Utilization, Northeast Institute of Geography and Agroecology, Chinese Academy of Sciences, Harbin, China
[2] State Key Laboratory of Soil and Water Conservation and Desertification Control, Northwest A&F University, Yangling, China
[3] Institute of Soil and Water Conservation, Chinese Academy of Sciences and Ministry of Water Resources, Yangling, China
[4] Harbin Normal University, Harbin, China
[5] University of Chinese Academy of Sciences, Beijing, China

*Correspondence to*: Mingming Guo (guomingming@iga.ac.cn)

**Abstract.** Tracking the transport of runoff-dissolved nitrogen (N) and phosphorus (P) from upslope farmland to the catchment outlet is vital for controlling non-point source pollution in agroecosystems. However, the hydrological and regulatory roles of permanent gully within catchment in modulating dissolved N and P losses dynamics under natural rainfall conditions remain poorly understood. In this study, runoff and associated losses of dissolved $NH_4^+$, $NO_3^-$, and P were measured at both the gully head and the outlet from 2022 to 2023. The results are as follows: (1) Gully significantly enhanced runoff generation, contributing 36.1% of total runoff despite occupying only 12.4% of the area. This contribution varied across rainfall types (Type A, frequent, low-depth, low erosivity; Type B, short duration, high intensity; Type C, long duration, high erosivity) and was highest under Type A (43.2%) and lowest under Type C (33.8%). (2) Gully exerted a pronounced dilution effect on the concentrations of dissolved $NH_4^+$, $NO_3^-$, and P, particularly for dissolved $NO_3^-$ (dilution ratio: 0.65). Consequently, gully contributed less to dissolved nitrogen and phosphorus fluxes relative to its contribution to runoff volume, accounting for 31.4%, 22.4%, and 31.1% of dissolved $NH_4^+$, $NO_3^-$, and P fluxes, respectively. (3) Type C rainfall dominated the loss of dissolved N and P. Only 10.2% of events contributed over 68% of dissolved N and P fluxes at the catchment scale and markedly increased their loss sensitivity to rainfall compared to Type A and Type B. These sensitivities were also intensified by gully. The study provides new insights into runoff dissolved nutrient interactions within gully systems and offers a foundation for improving nutrient management in gully-dominated agricultural landscapes.



## 1 Introduction

The loss of nitrogen (N) and phosphorus (P) via agricultural surface runoff poses a major challenge to watershed management, as these nutrients are key contributors to downstream eutrophication (Berretta & Sansalone, 2011; McDowell & Haygarth, 2024; Huo et al., 2025). Dissolved nitrogen (DN) and dissolved phosphorus (DP), being the most mobile and bioavailable forms, are rapidly transported to aquatic systems during rainfall events, where they can trigger algal blooms due to their high ecological reactivity (Wang et al., 2024; Xiao et al., 2024). Compared to particulate forms, DN and DP respond more quickly to storm-driven hydrological processes and are more easily mobilized along surface flow paths (Berretta & Sansalone, 2011). In agricultural landscapes, these flow paths are often shaped by permanent gullies that act as hydrological conduits linking farmland with downstream water bodies. Gullies are widespread in farmland across China, the United States, and various regions of Europe and Australia (Dube et al., 2020; Walker et al., 2024; Chen et al., 2025c). However, their role in regulating the hydrological and dissolved nutrient dynamics of dissolved nutrient transport under natural rainfall remains insufficiently quantified.

Permanent gullies are geomorphic features formed through prolonged water erosion, serving as critical pathways that connect upslope farmland with downstream aquatic systems. Unlike engineered drainage ditches, these gullies typically lack vegetation cover, experience minimal human intervention, and are often subject to severe erosion (Wang et al., 2019; Kumar Bhattacharya et al., 2024). Such characteristics suggest that gullies function not only as efficient hydrological pathways but may also exhibit multifaceted roles in nutrient dynamics (as sources, sinks, or regulators) depending on prevailing hydrological conditions (He et al., 2024). DN and DP, owing to their higher mobility and bioavailability, are more responsive to hydrological processes and land use changes than their particle forms (Lee et al., 2013). Land use exerts a critical influence on nutrient fluxes: forests, grasslands, and riparian buffers often act as nutrient sinks (Räty et al., 2020), whereas intensively managed croplands, frequently subject to fertilizer misapplication, represent major nutrient sources (Liu et al., 2020; Risal et al., 2020; Wang et al., 2025). In agricultural catchments, gullies predominantly receive runoff from upslope cultivated fields (Zhang et al., 2011), and their sparse vegetation and limited internal nutrient inputs inevitably modulate nutrient transport processes (Ezzati et al., 2020). Steep gully gradients further intensify runoff energy and hydrological connectivity, accelerating sediment transport (Kumar Bhattacharya et al., 2024). During rainfall events, deposited sediments within gullies may re-mobilize nutrients, presenting a potential risk of secondary pollution (Ezzati et al., 2020; Xu et al., 2022). Notably, such regulatory mechanisms are likely to vary with rainfall type. Rainfall characteristics, including depth, intensity, duration, and erosivity, are key drivers of runoff generation, erosion, and nutrient mobility in agricultural landscapes (Wang et al., 2024; Wang et al., 2025). Different rainfall types, such as high-frequency low-intensity versus low-frequency high-intensity events, can lead to substantial differences in nutrient transport pathways, efficiencies, and associated risks (Wang et al., 2024; Yang et al., 2024; Wang et al., 2025). With the growing intensity of extreme weather events under global climate change, heavy storms are increasingly associated with intense erosion and elevated nutrient loads, often resulting in dissolved N and P exports that greatly exceed those under moderate rainfall. Conversely, small



rainfall events may favor nutrient dilution or retention due to reduced flow velocities or altered concentration gradients (Wang et al., 2025). Disparities in soil properties, vegetation cover, and topography between slopes and gullies may further amplify these effects. Nevertheless, the role of gullies in modulating dissolved nutrient losses under varying rainfall conditions remains insufficiently investigated. A comprehensive understanding of their regulatory function is thus crucial for environmental sustainability at the watershed scale.

The Mollisols region of Northeast China (MRNC) produces approximately 50% of China's rice, 44% of its soybeans, and 34% of its corn, thus serving as a cornerstone of national food security (Chen et al., 2025a). However, decades of extensive and intensive land development have resulted in widespread gully erosion and land degradation, rendering the region increasingly vulnerable to ecological stress. To date, over 667,000 permanent gullies have been identified, with more than 85% remaining active, posing serious threats to agricultural sustainability and watershed integrity (Chen et al., 2025c). Earlier studies have explored the influence of rainfall characteristics on gully formation (Tang et al., 2023; Liu et al., 2024), as well as the function of gullies in sediment and nutrient losses during snowmelt events (Su et al., 2024), limited attention has been paid to the regulatory function of permanent gullies in DN and DP transport under natural rainfall conditions. This knowledge gap is largely attributed to technical challenges in field-based monitoring, which have constrained a comprehensive understanding of gully-mediated nutrient dynamics and their implications for watershed-scale water quality management in the MRNC.

To address these gaps, this study conducted in situ monitoring of runoff and associated losses of dissolved $NH_4^+$, $NO_3^-$, and P at both the gully head and outlet in two agricultural catchments in MRNC during natural rainfall in 2022 and 2023. The specific objectives were to: (1) elucidate the regulatory effect of gully to runoff, dissolved $NH_4^+$, $NO_3^-$, and P losses; (2) quantify how gully contributions to these losses vary in response to different rainfall types; and (3) reveal how gully regulate the response relationship between rainfall and dissolved $NH_4^+$, $NO_3^-$, and P losses. The findings provide a scientific basis for non-point source pollution mitigation and control.

## 2 Materials and methods

### 2.1 Study area

(1) The study area is located in Guangrong Village (N 47°34′-47°38′, E 126°81′-126°88′), Hailun City, Heilongjiang Province, within the central MRNC (Fig. 1A). The region experiences a continental monsoon climate, with annual precipitation of 300–900 mm (2000–2022), ~80.7% of which falls between June and October, coinciding with peak soil erosion. Mean annual temperature is ~1.5°C (-25.6°C to 26.6°C), with crop sowing typically commencing in mid-April. The terrain comprises gently rolling hills, and soils are classified as Mollisols (Chernozem) with silty clay loam texture, 45–60% silt content, and >3% organic matter. These conditions support intensive maize and soybean cultivation, but sustained anthropogenic disturbance has caused a ~20% decline in soil fertility. Especially, gully erosion on sloping farmland leads to an annual arable land loss of ~0.097%, with gully density reaching 1.5 km km$^{-2}$ (Chen et al., 2025c).





(2) During rainfall events, these gullies serve as efficient conduits for runoff generated from upslope farmlands. To elucidate the role that gullies play in mediating the transport of dissolved nitrogen and phosphorus within this runoff, a comprehensive gully survey was conducted in May 2021 prior to hydrological monitoring. This survey aimed to assess the morphological characteristics and activity status of the gullies in the region, thereby enabling the selection of representative gullies for detailed investigation. The results revealed that over 90% of farmland gullies were found to be highly active, with

average widths and depths of 13.3 m and 3.4 m, respectively.

    (3) On this basis, two representative and actively eroding farmland gullies (F1 and F2; the two catchments are 1.1 km apart) were selected (Fig. 1B–E), exhibiting similar catchment areas, land use proportions, and typical morphological and topographic features. The F1 and F2 catchments cover 4.3 ha and 3.4 ha, respectively. Farmland is the dominant land use, comprising 83.4% of F1 and 85.5% of F2. Both gullies within catchment showed pronounced erosion, including active

headcuts and exposed sidewalls. Gully dimensions were consistent with survey averages: F1 measured 0.38 ha (area), 242.3 m (length), 17.7 m (width), and 3.8 m (depth); F2 measured 0.54 ha, 293.7 m, 18.4 m, and 4.8 m, respectively. Gully slope gradients (F1: 36.2°; F2: 39.5°) were significantly steeper than adjacent farmland slopes (F1: 4.3°; F2: 3.4°). A 2 m wide unplanted buffer zone, maintained along gully banks for machinery access, was colonized by natural grass cover. Field monitoring during intense rainfall indicated that these grass strips, together with wheel tracks, effectively diverted lateral

runoff downslope along their margins, reducing direct flow into the gullies (Chen et al., 2025b).

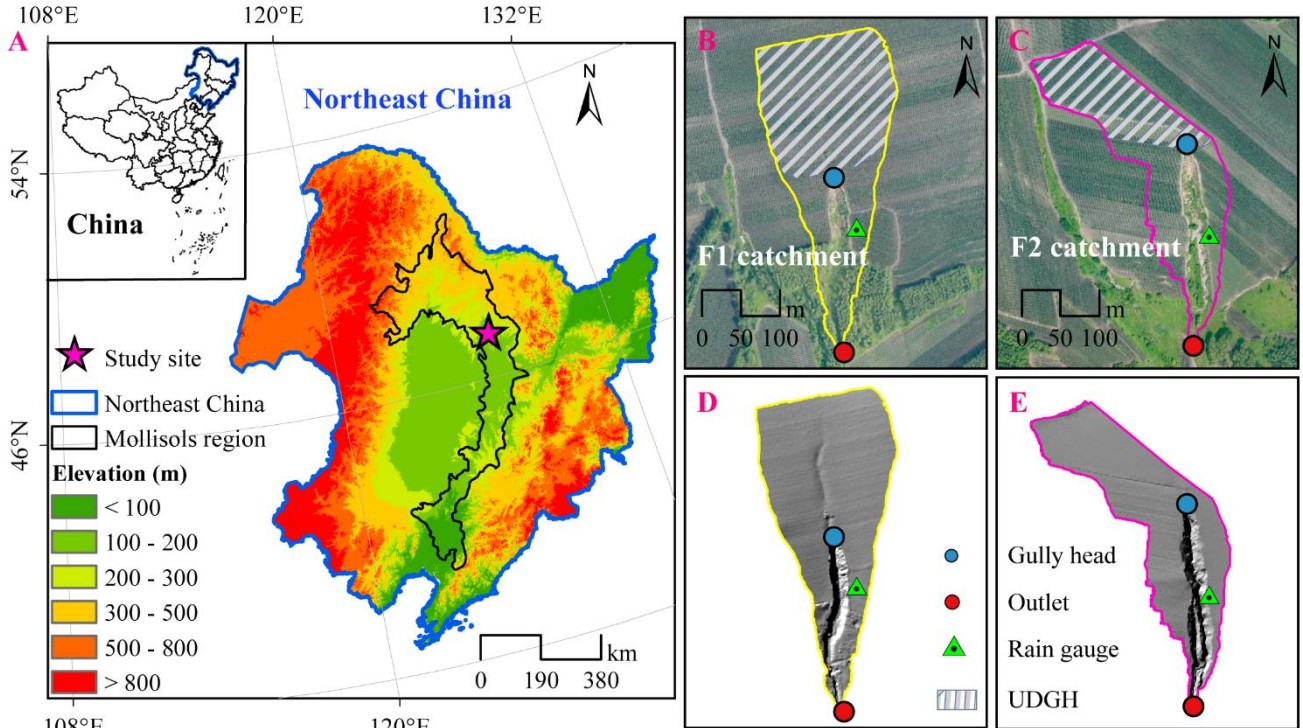

**Fig. 1 (A) Location of the study site within MRNC; (B–E) overview of the two monitored gully-dominated catchments. UDGH represents the upslope drainage area of gully head.**



## 2.2 Rainfall data capture

From June to October in both 2022 and 2023, rain gauges with a resolution of 0.2 mm were installed in both catchments to record spatially heterogeneous rainfall (Fig. 1B–C). A rainfall event was characterized as a continuous period of precipitation separated by no more than 6 hours; longer intervals were treated as separate events. Rainfall events that triggered significant soil erosion were classified as erosive rainfall (Chen et al., 2024). To evaluate the impacts of different rainfall types on dissolved $NH_4^+$, $NO_3^-$, and P losses, five parameters were selected for cluster analysis: rainfall depth,

duration, average intensity, maximum 30-minute intensity, and rainfall erosivity. The calculation of rainfall erosivity (RE) is provided in Equations (1)–(3):

$$RE = K_e \cdot I_{30} \tag{1}$$

$$K_e = \sum(E_r \cdot P) \tag{2}$$

$$E_r = 0.29[1 - 0.72e^{-0.082i_r t}] \tag{3}$$

In the equation, $K_e$ represents the total kinetic energy of a rainfall event (MJ hm$^{-2}$); P is the total rainfall amount during the event (mm); Er denotes the unit kinetic energy during sub-event r (MJ hm$^{-2}$ mm$^{-1}$); R = 1, 2, ..., n refers to the sub-events into which a single rainfall event is divided based on rainfall intensity; and $i_r$ is the rainfall intensity during sub-event r (mm h$^{-1}$).

## 2.3 Runoff monitoring and sample processing

To examine how dissolved $NH_4^+$, $NO_3^-$ and P transport in runoff changes after entering the gully, V-shaped weirs were installed at both the gully head (upslope drainage area of the gully head; UDGH) and the catchment outlet. HOBO Water Level Probes recorded runoff dynamics at 10-minute intervals by measuring pressure differences relative to identical probes placed in the air (Cheng et al., 2023; Chen et al., 2025b). Runoff samples were manually collected during rainfall events at pre-, mid-, and post-runoff stages using 1,000 mL polyethylene bottles. The sampling frequency was dynamically adjusted

based on runoff fluctuations to capture peak flow dynamics. During rapid discharge changes, intervals were shortened to as little as 2 min. As flow subsided, the sampling frequency was reduced accordingly. Once sufficient samples had been obtained to characterize the runoff process, intervals were further extended to ensure full coverage of the event. All collected samples were immediately delivered to the laboratory for dissolved $NH_4^+$, $NO_3^-$, and P analysis.

    A subsample was filtered through a 0.45 μm Millipore membrane to obtain the filtrate for nutrient analysis.

Concentrations of dissolved $NH_4^+$, $NO_3^-$, and P were determined using standard spectrophotometric methods: Nessler's reagent spectrophotometry for $NH_4^+$, ultraviolet spectrophotometry for $NO_3^-$, and ammonium molybdate spectrophotometry for DP. The runoff volume for each rainfall event was calculated using the calibrated flume depth-runoff curve and an empirical formula. By integrating high-frequency runoff sampling and dissolved nutrient concentrations, the dissolved nutrient load induced by each rainfall event was determined. A detailed description of the calculation process can be found in

our previous research (Chen et al., 2025b).



## 2.4 Data analysis

Rainfall types were classified using K-means clustering analysis via the R package "cluster" (v.2.1.3). Data normality and variance homogeneity were assessed using Shapiro's test and Levene's test, respectively. If assumptions were met, one-way ANOVA followed by Tukey's HSD test was used to compare dissolved $NH_4^+$, $NO_3^-$, and P loss across rainfall types; otherwise, the Kruskal-Wallis nonparametric test was applied. A statistically significant difference ($P < 0.05$) was interpreted as evidence of gully-mediated regulation of nutrient export dynamics under different rainfall types. Redundancy analysis was employed to explore the relationships between rainfall, runoff, dissolved $NH_4^+$, $NO_3^-$, and P losses. All statistical analyses were performed in R (v.4.5.0).

## 3 Results

### 3.1 Rainfall characteristics

From June to October 2022 and 2023, 30 and 29 rainfall events were recorded in F1 and F2, respectively. K-means clustering classified these events into three distinct rainfall types (Table 1). Type A was dominant, occurring 23 times in both catchments, while Types B and C were less frequent (F1: 4 and 3; F2: 3 and 3, respectively). Type A was characterized by low rainfall depth (20.8 mm), moderate duration (491.6 min), moderate intensity (3.9 mm h⁻¹), and low erosivity (71.9 MJ mm ha⁻¹ h⁻¹). Type B featured moderate depth (23.8 mm), short duration (56.1 min), high intensity (28.3 mm h⁻¹), and moderate erosivity (267.4 MJ mm ha⁻¹ h⁻¹). Type C exhibited high depth (79.6 mm), long duration (2922.4 min), low intensity (1.7 mm h⁻¹), and the highest erosivity (333.7 MJ mm ha⁻¹ h⁻¹). Despite their lower frequency, the erosivity of Types B and C was 3.6 and 4.3 times higher than that of Type A, respectively.

**Table 1 Average values of rainfall parameters for the three rainfall patterns**

| | Rainfall type | Sample size | P (mm) | D (min) | $I_{mean}$ (mm h⁻¹) | $I_{30}$ (mm h⁻¹) | RE (MJ mm hm⁻² h⁻¹) |
|------|------|------|------|------|------|------|------|
| F1 | A | 23 | 21.5 | 525.1 | 3.4 | 17.1 | 84.5 |
| | B | 4 | 24.5 | 52.8 | 30.3 | 46.1 | 334.2 |
| | C | 3 | 82.5 | 2959.7 | 1.8 | 20.8 | 515.7 |
| F2 | A | 23 | 20.1 | 458.2 | 4.5 | 13.7 | 59.3 |
| | B | 3 | 23.1 | 59.3 | 26.2 | 32.9 | 200.6 |
| | C | 3 | 76.7 | 2885.0 | 1.6 | 10.5 | 151.6 |

Abbreviations: P, Rainfall depth; D, Rainfall duration; $I_{mean}$, Mean rainfall intensity; $I_{30}$, maximum 30-min rainfall intensity; RE, rainfall erosivity.



## 3.2 The role of gully in regulating runoff

During Type C rainfall, cumulative runoff volume in UDGH was 3.9 and 21.0 times greater than that under Types A and B, respectively. At the outlet, cumulative runoff under Type C was 3.3 times higher than under Type A and 19.0 times higher than under Type B. On average, Type C rainfall generated significantly more runoff than Types A and B at both locations ($P < 0.05$). Specifically, the average runoff volume in the UDGH during Type C was 29.8 and 24.5 times greater than under Types A and B, respectively, while at the outlet, it was 25.6 and 22.1 times higher (Fig. 2A–B). Although Type B produced more runoff than Type A, the difference was not statistically significant ($P > 0.05$) (Fig. 2A–B).

Gully accounted for only 12.4% of the catchment area but contributed an average of 36.1% of total runoff (based on the mean of catchments F1 and F2). This contribution varied with rainfall type, with the highest observed under Type A (43.2%), followed by Type B (40.1%), and the lowest under Type C (33.8%) (Fig. 2C–D).



**Fig. 2 (A–B) Proportion of the accumulated runoff loss on UDGH and gully. (C–D) Impact of different rainfall types on runoff volume. Bars without filled colors represent cumulative runoff volume under different rainfall types, embedded bars with filled colors represent the average runoff volume for individual rainfall events, and different lowercase letters represent significant differences in runoff volume between different rainfall types. Abbreviation: UDGH represents the upslope drainage area of gully head.**

### 3.3 The transport effect of dissolved $NH_4^+$, $NO_3^-$ and P by gully

#### 3.3.1 Dissolved $NH_4^+$, $NO_3^-$ and P concentration

Dissolved $NH_4^+$, $NO_3^-$ and P concentrations measured at the outlet were consistently lower than those observed at the gully head. On average, the dilution ratios (outlet concentration divided by gully head concentration) were 0.77 for dissolved $NH_4^+$, 0.65 for $NO_3^-$, and 0.87 for P. These values indicated that the gully exerted a stronger dilution effect on $NO_3^-$ than on $NH_4^+$ or P (Fig. 3).

The effect of gully on dissolved $NH_4^+$, $NO_3^-$ and P concentrations also varied with rainfall type (Fig. 4). On average, dissolved $NH_4^+$ concentrations at the gully head were 1.33, 1.24, and 1.21 times higher than those at the outlet under rainfall Types A, B, and C, respectively. For dissolved $NO_3^-$, the corresponding ratios were 1.61, 1.58, and 1.21. For DP, the ratios were 1.19, 0.94, and 1.21. These results suggested that under rainfall Types A and B, gully intensified the concentration gradient of $NH_4^+$ and $NO_3^-$ between the gully head and outlet. In contrast, the pattern for DP appeared more variable: strong dilution occurred under Types A and C, whereas slight increases were observed under Types B, possibly indicating episodic in-channel P release.

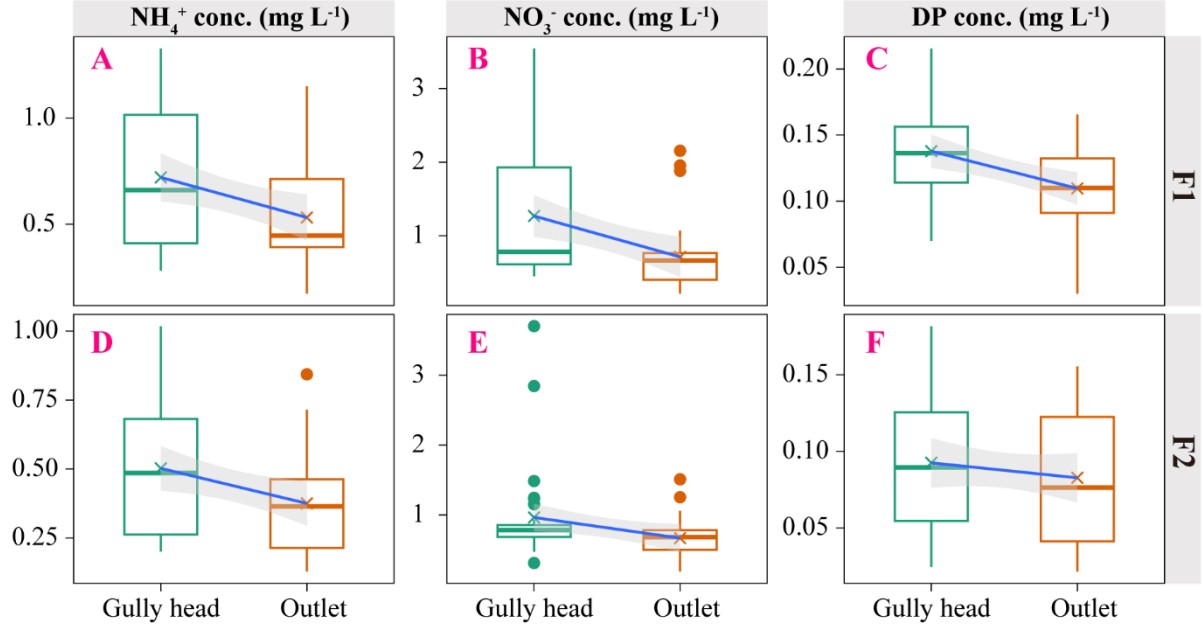

**Fig. 3 Comparison of dissolved $NH_4^+$, $NO_3^-$ and P concentrations at gully head and outlet.**





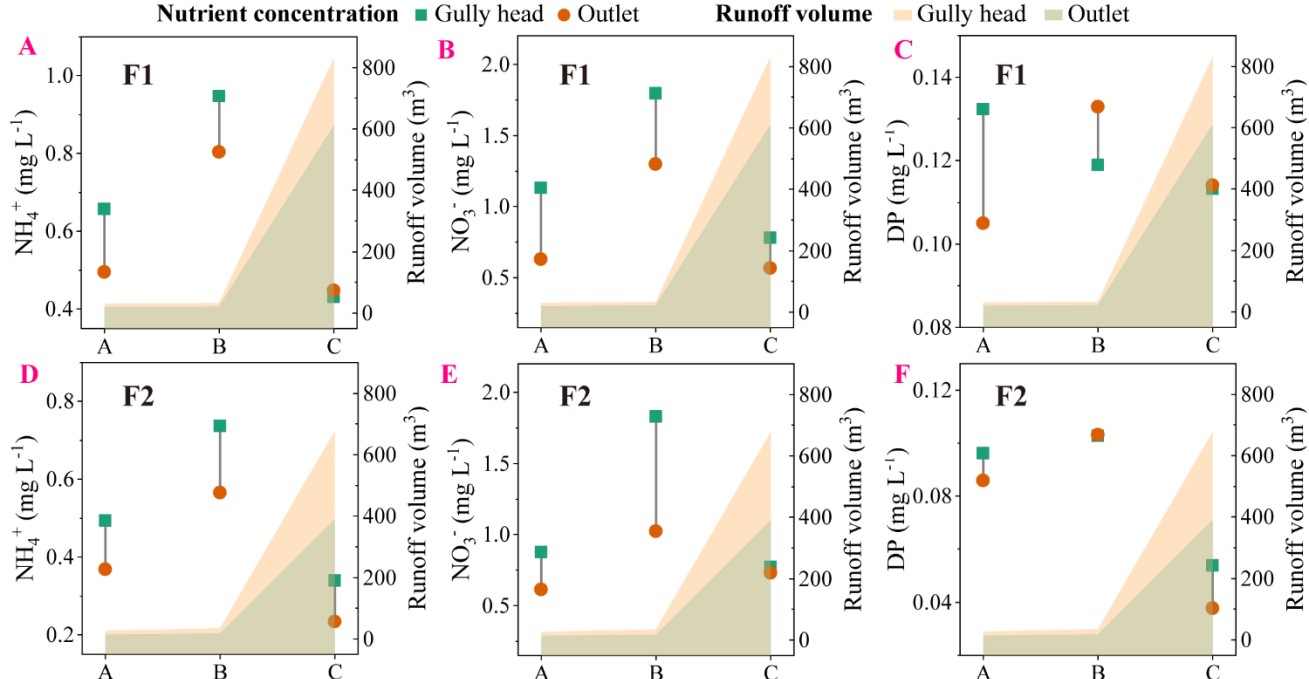

**Fig. 4** Characterization of dissolved $NH_4^+$, $NO_3^-$ and P concentrations under different rainfall types.

### 3.3.2 Transport flux of dissolved $NH_4^+$, $NO_3^-$ and P

Gully contributed 31.4%, 22.4%, and 31.1% of the total dissolved $NH_4^+$, $NO_3^-$, and P transport fluxes at the catchment scale, respectively (Fig. 5).

Rainfall type had a significant impact on dissolved nutrient transport. Although Type C rainfall accounted for only 10.2% of events, it contributed 78.1%, 73.4%, and 71.9% of the gully transport fluxes of $NH_4^+$, $NO_3^-$, and P, respectively. At the catchment scale, Type C events similarly dominated, contributing 68.2%, 73.8%, and 71.8% of the total fluxes. On average, gully accounted for 27.1%, 15.3%, and 34.5% of $NH_4^+$ transport under Types A, B, and C, and 24.8%, 8.0%, and 23.2% of $NO_3^-$ transport, respectively. These results indicated the strongest reduction of $NH_4^+$ and $NO_3^-$ fluxes under Type B rainfall and the weakest under Type C. In contrast, gully contributions to DP transport were 22.7%, 40.9%, and 33.1% under Types A, B, and C, respectively, suggesting a reduced regulatory effect during Type B events and an enhanced effect during Type A (Fig. 6).

At the event scale, transport fluxes of dissolved $NH_4^+$, $NO_3^-$, and P were significantly higher under Type C rainfall compared to Types A and B ($P < 0.05$). Although Type B fluxes exceeded those of Type A, the differences were not statistically significant ($P > 0.05$) (Fig. 7).



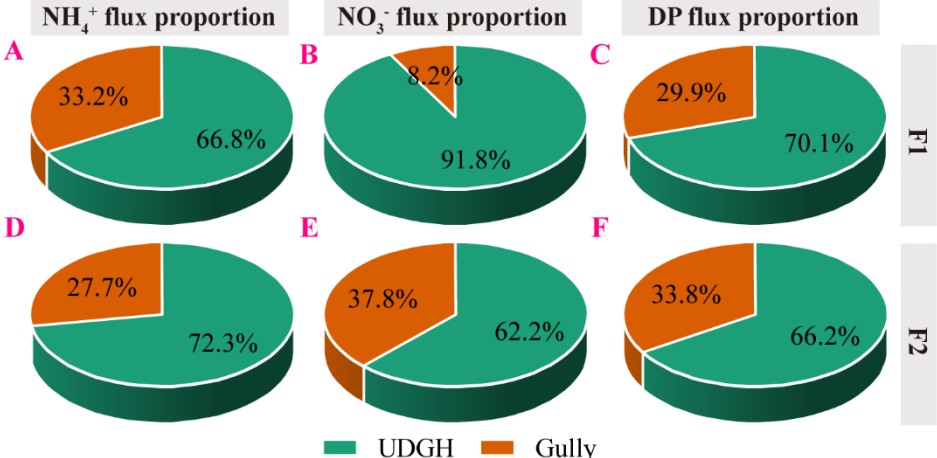

**Fig. 5 Contribution of different sites to dissolved NH$_4^+$, NO$_3^-$ and P loss flux in agricultural catchments. Abbreviation: UDGH represents the upslope drainage area of gully head.**

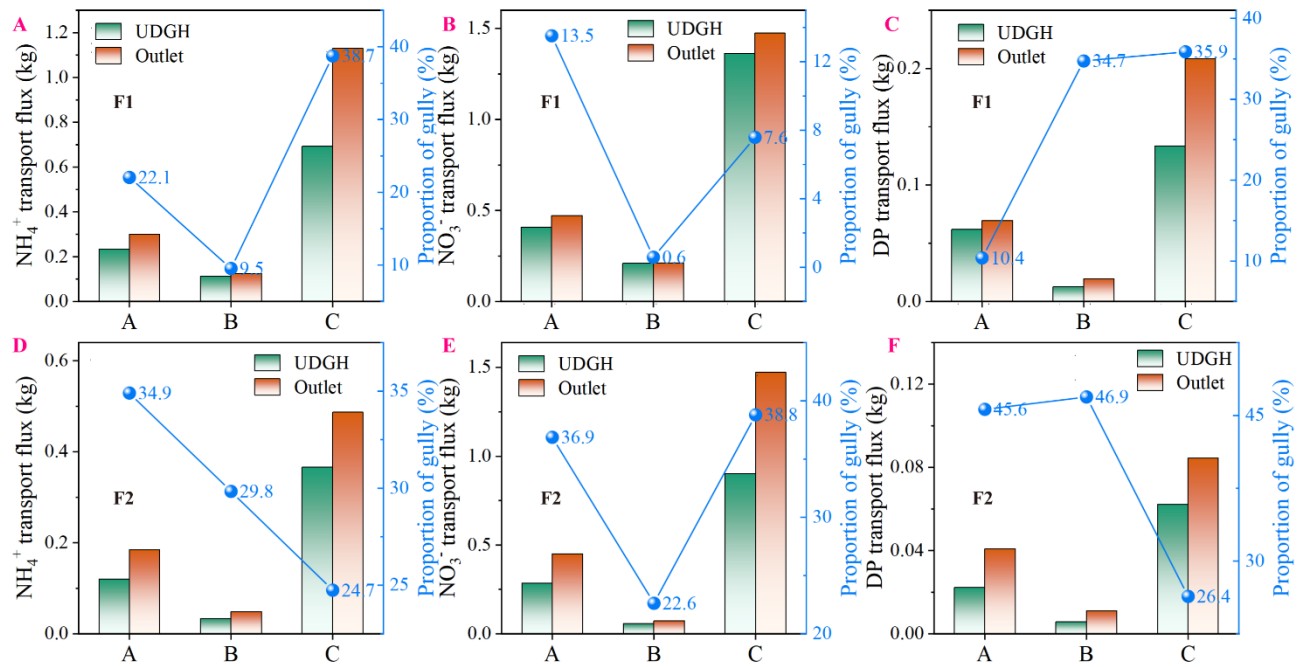

**Fig. 6 Characterization of dissolved NH$_4^+$, NO$_3^-$ and P cumulative transport flux under different rainfall types.**



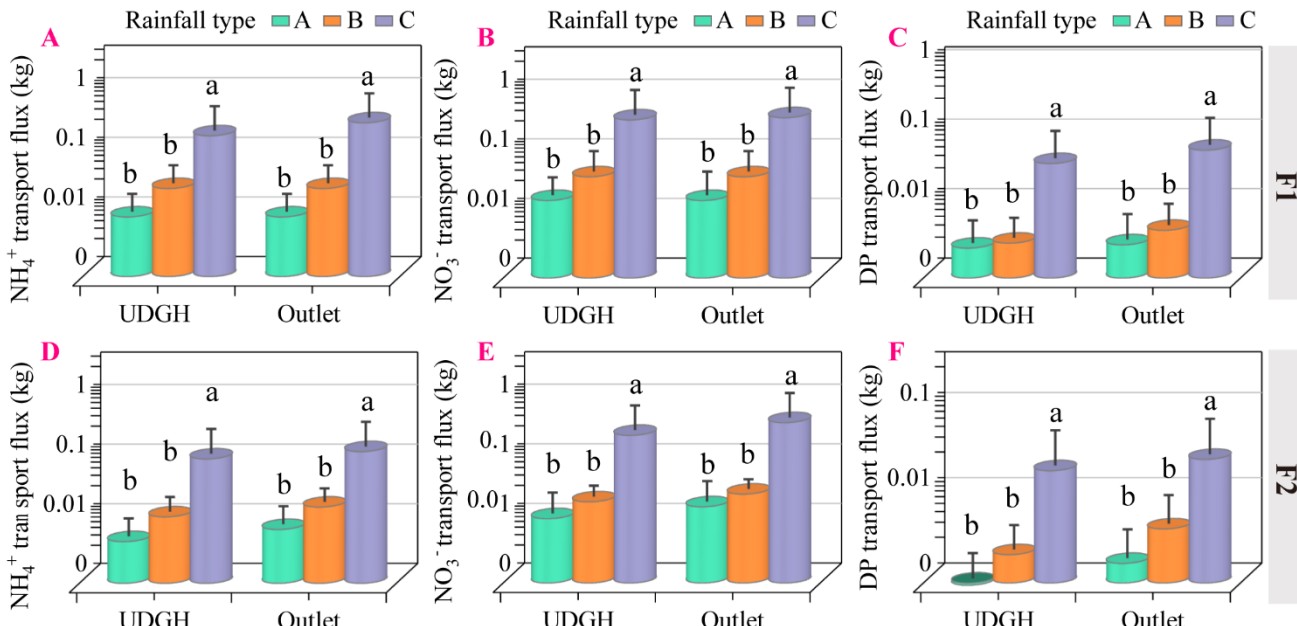

**Fig. 7 Effect of different rainfall types on the average transport flux of dissolved NH$_4^+$, NO$_3^-$ and P at event scale.**

## 3.4 Gully regulates response relationships among runoff and dissolved NH$_4^+$, NO$_3^-$ and P fluxes

### 3.4.1 Correlation analysis of rainfall factors with runoff and dissolved NH$_4^+$, NO$_3^-$ and P fluxes

At both the gully head and outlet, rainfall depth was the factor with the greatest correlation with runoff volume and dissolved NH$_4^+$, NO$_3^-$, and P fluxes, followed by rainfall erosivity. Relative to the gully head (IUGH), the presence of the gully led to only a slight enhancement in the correlation between runoff volume, dissolved NH$_4^+$, NO$_3^-$, and P fluxes and rainfall depth at the outlet (Fig. 8).



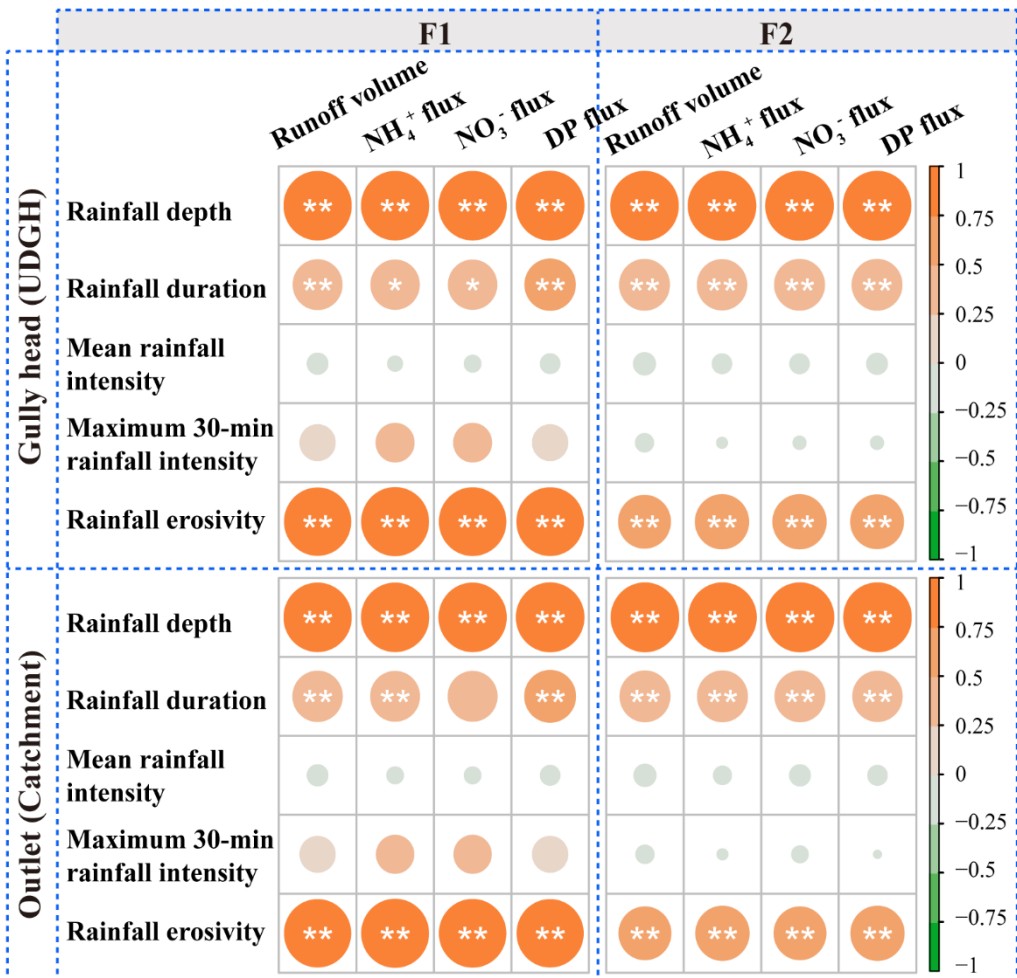

230

**Fig. 8 Correlation analysis of rainfall characteristics with runoff and dissolved NH$_4^+$, NO$_3^-$ and P fluxes.**

**3.4.2 Response of dissolved NH$_4^+$, NO$_3^-$ and P transport flux to rainfall, runoff, and concentration**

Redundancy analysis was used to reveal dominant factors of dissolved nutrient fluxes, and the results indicated that the transport fluxes of dissolved NH$_4^+$, NO$_3^-$, and P fluxes were primarily influenced by runoff volume, rainfall depth, and

235   rainfall type, while their correlations with corresponding concentrations were not significant. This indicated that the transport of dissolved NH$_4^+$, NO$_3^-$, and P was mainly controlled by runoff rather than concentration (Fig. 9).

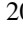




**Fig. 9 Ordination plots of redundancy analysis for the dissolved NH$_4^+$, NO$_3^-$ and P transport flux response. Variation partitioning analysis of rainfall, runoff, and dissolved nutrient concentration to variation in dissolved nutrient transport flux and individual effects of explanatory variables based on hierarchical partitioning.**

### 3.4.3 Gully modulated the response relationship between rainfall, dissolved NH$_4^+$, NO$_3^-$, and P fluxes

A significant power function relationship ($F=aR^b$) was observed between rainfall depth and the transport fluxes of dissolved NH$_4^+$, NO$_3^-$, and P, where the coefficient a indicates the sensitivity of nutrient loss to rainfall (higher values reflect greater mobilization potential) and the exponent b represents the transport efficiency at which fluxes respond to changes in rainfall depth (P < 0.01; Fig. 10). The results revealed that gully significantly increased the sensitivity coefficient (a) for




dissolved $NH_4^+$, $NO_3^-$, and P transport fluxes. However, gully also reduced the overall transport efficiency (Fig. 10). Among the different rainfall types, Type C rainfall markedly enhanced the sensitivity of dissolved $NH_4^+$, $NO_3^-$, and P fluxes compared to Types A and B. Furthermore, within the same rainfall category, a comparison of slope values of the linear equation between the gully head and the outlet showed that the presence of the gully amplified the rainfall sensitivity of

250   dissolved $NH_4^+$, $NO_3^-$, and P transport at the catchment outlet (Fig. 11).

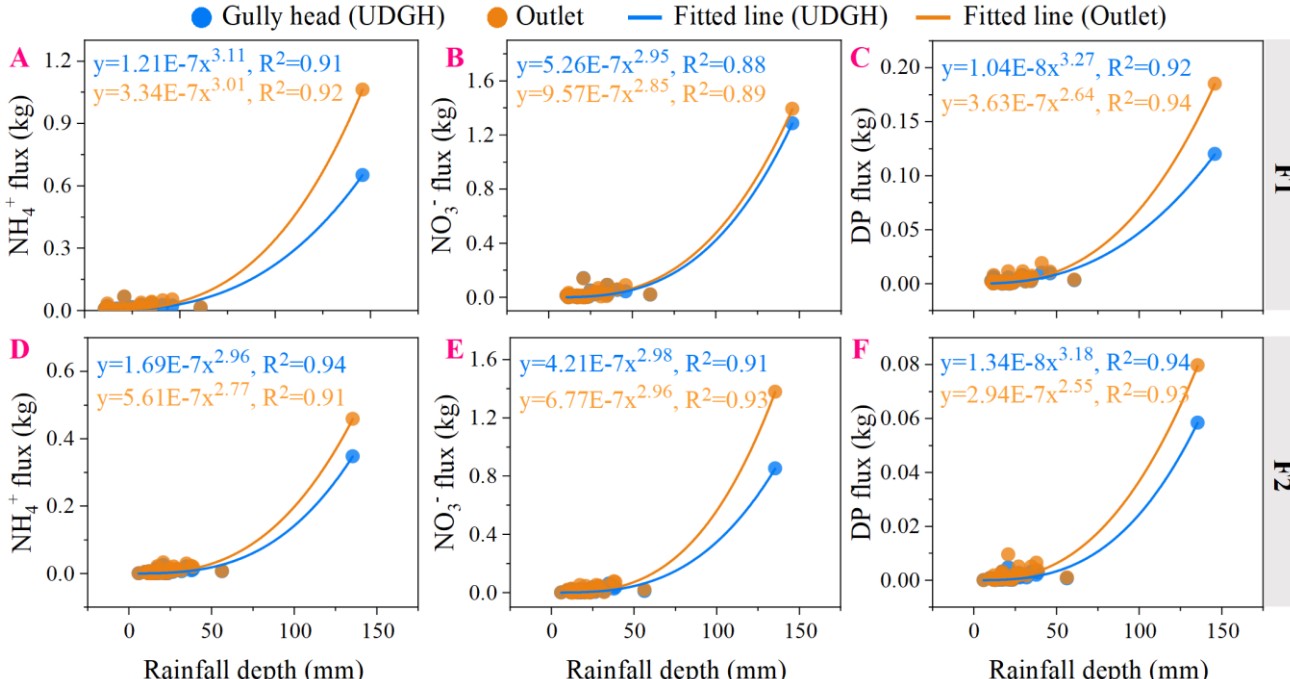

**Fig. 10 Differences in the response of dissolved $NH_4^+$, $NO_3^-$, and P transport fluxes to runoff volume at gully head and outlet. Solid lines indicate standard major axis (SMA) regression lines. P < 0.05 indicates a significant difference between the slopes of the regression lines at the gully head and outlet.**

255





**Fig. 11 Differences in the response of dissolved NH$_4^+$, NO$_3^-$, and P transport fluxes to rainfall depth under different rainfall types.**

## 4 Discussion

### 4.1 Regulating effect of gully on transport of dissolved NH$_4^+$, NO$_3^-$, and P

260    Our findings indicated that gully, despite occupying only 12.4% of the catchment area, contributed 36.1% of the total runoff (Fig. 2). Compared with the gently sloping farmland covered by dense crops, the steep topography and sparse




vegetation of gully provided favorable conditions for the generation and concentration of runoff, which subsequently transported dissolved nutrients downstream to rivers and lakes (Hou et al., 2022; Chen et al., 2024; Chen et al., 2025b). Notably, gully exerted a strong dilution effect on dissolved nutrients, especially $NO_3^-$, with an average concentration ratio of 265 0.65 between the outlet and the gully head (Fig. 3). This pronounced reduction in runoff $NO_3^-$ concentration may have resulted from the formation of ponded, anaerobic, or reducing microenvironments in locally flat sections of the gully bed, where denitrifying microorganisms could convert $NO_3^-$ into gaseous nitrogen, thereby significantly lowering its concentration. Furthermore, runoff $NO_3^-$, as a highly mobile anion, is not readily adsorbed by sediments and tends to distribute evenly in gully water. As runoff accumulated, $NO_3^-$ was more prone to dilution than retention (Wang et al., 2024; 270 Zhao et al., 2025). In contrast, $NH_4^+$, as a positively charged ion, is more likely to be adsorbed and immobilized by sediments on the gully bed, particularly under fluctuating hydrodynamic conditions such as floods (Zhao et al., 2025). These conditions are common in gully, which were globally recognized as major sediment sources (Kumar Bhattacharya et al., 2024; Su et al., 2024; Chen et al., 2025b). Therefore, the observed reduction in $NH_4^+$ concentrations may have been more the result of physical retention than dilution (Wang et al., 2024). P, unlike N, does not undergo gaseous transformation and is 275 primarily governed by precipitation and adsorption processes, making its dilution response less straightforward (Liu et al., 2020; Yang et al., 2024). Interestingly, while gully generally reduced the concentrations of dissolved $NH_4^+$, $NO_3^-$, and P from upslope runoff, they also amplified the sensitivity of transport fluxes to runoff (Fig. 10). In other words, the transport of dissolved $NH_4^+$, $NO_3^-$, and P was primarily governed by runoff volume rather than concentration (Fig. 9). Even a small increase in runoff led to a disproportionate surge in nutrient export, potentially offsetting the dilution or interception effects 280 of gullies. Therefore, effective regulation of runoff distribution within catchments may play a critical role in mitigating dissolved nutrient losses.

**4.2 Regulation effect of gully was influenced by rainfall types**

In this study, concentrations of dissolved $NH_4^+$, $NO_3^-$, and P were significantly higher during light rainfall events compared to rainstorms (Fig. 4). Similar trends have been observed in Southwest China, where nutrient concentrations in 285 runoff decreased with increasing rainfall following straw return practices on sloping farmland (Zhang et al., 2024; Feng et al., 2025). In contrast, monitoring in micro-catchments comprising paddy fields and drylands found that peak concentrations of dissolved $NH_4^+$ and $NO_3^-$ followed the order: heavy rainstorm event > rainstorm > moderate rain (Zhang et al., 2011). These contrasting patterns may reflect differences in runoff volume and land use. Rainstorms can generate large runoff volumes that dilute nutrient concentrations (Wenng et al., 2020). However, nutrient concentrations are also influenced by source 290 conditions. Fertilization on slopes replenishes surface soil nutrients, and if followed by heavy rainfall, may lead to elevated nutrient losses compared to less intense events (Liu et al., 2020; Wenng et al., 2020). This highlights the importance of managing the timing of fertilizer application, especially before extreme rainfall. Meanwhile, our results also indicated that the regulatory effect of the gully on dissolved $NH_4^+$ and $NO_3^-$ concentrations weakens under extreme rainfall, while its influence on DP concentrations increases (Fig. 4; Fig. 6). This may be attributed to the distinct environmental conditions





within gully compared to upslope farmland. Gully soils are typically less fertile and less responsive to nutrient mobilization by rainfall. In addition, steep gully slopes promote rapid runoff from upslope farmland, especially during extreme events, when high rainfall intensity and long duration accelerate soil saturation and shorten hydrological response times, thereby enhancing the gully's regulating role (Wenng et al., 2020; Wang et al., 2025). As discussed earlier (Section 4.1), the mechanisms driving DN and DP losses differ. Under varying rainfall conditions, the heterogeneity in gully soil, topography,

and vegetation may intensify these differences, leading to inconsistent patterns of nutrient concentrations across rainfall types (Feng et al., 2025). These findings emphasize the need for rainfall-specific management strategies in agricultural catchments.

### 4.3 Implications for agricultural catchment management

This study demonstrates that under natural rainfall conditions, the gully in agricultural catchments plays a dual role.

The runoff-amplifying function of the gully makes it a major contributor to total catchment runoff, especially during frequent, low-intensity rainfall events. At the same time, its dilution effect on dissolved nutrient concentrations partially offsets the increased nutrient fluxes associated with enhanced runoff. Notably, the regulatory effect on dissolved nutrients suggested that the gully may, under certain conditions, function as a buffer mitigating soluble nutrient losses (Krzeminska et al., 2023). These findings challenge the conventional view of the gully as a passive conduit of non-point source pollution. Instead, its

response and regulatory capacity vary significantly with the rainfall type. Effective management should therefore account for the gully's developmental stage, spatial position, and hydrological–biogeochemical behavior. Measures such as vegetation restoration and landscape optimization may enhance gully function (Wang et al., 2023; Wu et al., 2025). For example, in regions with frequent small rainfall events, maintaining certain hydrological functions of the gully may be beneficial, whereas in areas prone to intense storms, enhancing its retention and interception capacity becomes essential.

Simultaneously, controlling nutrient sources remains critical. Practices such as straw return, balanced fertilization, and accurate field management can reduce nutrient concentrations and loss risk from upslope farmlands (Bayad et al., 2022; Chen et al., 2025a; Cui et al., 2025; Feng et al., 2025). An integrated management approach that combines field-scale practices with gully-scale interventions can help balance hydrological connectivity with nutrient loss control. This offers a practical foundation for promoting sustainable agriculture and protecting water quality in the Mollisol regions (Wenng et al.,

320   2020).

### 5 Conclusions

In this study, two gully-dominated small agricultural catchments in the Mollisols region of Northeast China were monitored under natural rainfall conditions from 2022 to 2023 to assess runoff and dissolved $NH_4^+$, $NO_3^-$, and P losses at both the upslope inflow (gully head) and the outlet. The key findings are as follows: Gully significantly amplified catchment

runoff generation. Despite occupying only 12.4% of the total catchment area, it contributed 36.1% of total runoff, rising to

43.2% under frequent, low-intensity, and low-erosivity rainfall events. After runoff entered the gully, notable dilution of dissolved $NH_4^+$, $NO_3^-$, and P concentrations occurred, with the strongest effect observed for $NO_3^-$. Due to dilution, the gully's contributions to dissolved $NH_4^+$, $NO_3^-$, and P fluxes were all lower than its runoff contribution. Low-frequency, high-erosivity rainfall events (10.2% of total events) dominated dissolved nutrient transport, accounting for 68.2%, 73.8%, and

71.8% of total $NH_4^+$, $NO_3^-$, and P fluxes, respectively. Dissolved nutrient fluxes were more strongly influenced by runoff volume than by changes in concentration, leading to increased sensitivity of dissolved nutrient transport to rainfall within the gully system. In summary, the developing gully functions not only as a hydrological conduit linking upslope farmland with downstream water bodies but also plays a regulatory role in dissolved N and P transport under variable rainfall types. These findings enhance understanding of non-point source pollution processes and provide a scientific foundation for targeted gully

management in the Mollisols region.

**Data availability**

Data will be made available upon request (guomingming@iga.ac.cn).

**Author contributions**

Z.C., M.G., and X.Z. conceived and designed the study. Z.C. prepared the initial manuscript, and M.G. and J.J.

contributed to manuscript revision. Z.C., M.G., L.W., X.L., and Q.C. were deeply involved in data collection and discussions on experimental design. M.G. and X.Z. provided financial support for the research.

**Financial support**

This study was supported by the Strategic Priority Research Program of the Chinese Academy of Sciences (XDA28010200), and Young Scientist Group Project of Northeast Institute of Geography and Agroecology, Chinese

Academy of Sciences (2023QNXZ03).

**Competing interests**

The authors declare that they have no conflict of interest.

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
