# Peer review of "Regulatory role of permanent gully in runoff dissolved nitrogen and phosphorus transport across rainfall types"

_EGUsphere, 2025_

## Referee Comment (RC1)

Overall summary: This study by Chen et al. aims to understand how gullies influence nutrient transport under different rainfall conditions. This research is ethically sound, scientifically valid, and technically accurate. Additionally, the methods are clear, making this reproducible. Overall, the paper is well-written and not too long, and the authors do a good job of describing why this research is important. I recommend professional English-language editing to improve clarity and consistency, especially in the abstract. For example, the term *gully* is repeatedly used as a plural noun; this should be corrected to *gullies* throughout the manuscript. Consequently, I have restricted my language-related comments and focus on the scientific content. Minor comments are listed below.

Minor Comments:

L17: Define ammonium and nitrate before acronym use.

L42: Perhaps a photo (could be supplemental) of a gully could be useful here.

L79-80: Define ammonium and nitrate before acronym use.

L118: How did you determine how much erosion occurred? Is this using the equation below, or did you ever measure suspended sediments? And what was the threshold for "significant" here?

L157-163: I recommend putting the sentences that describe the types of rain (A/B/C) first, followed by the type A was dominant, followed by B and C sentence.

Results (general): I am finding it difficult to remember the differences in the types of rainfall (A, B, C). Perhaps you can call them something different or redefine them in the captions of the figures?

Figures (general): In any figure with multiple panels, please define each letter in the caption. This is also a lot of figures, and a lot of information. Some of the figures only have a couple of sentences describing their results. I would recommend moving a couple of these to the supplemental, in order to make this an easier read and reduce redundancy.

Figure 11: Type C relationships seem to be driven by a single point. Did you check if this is an outlier? Also, these all have regression lines on them. Are they all significant relationships?

Discussion (general): I think the discussion could be a bit longer. There are so many results, and I think you could pull more from the literature to place this into context and provide suggestions for future research.

Conclusion: Most of the conclusion is just repeated from results. I think you could cut it down to just the last couple of sentences.